# Advances in Endoscopic Ultrasonography-Based Diagnosis of Pancreatic Lesions: Narrative Review

**DOI:** 10.3390/cancers17020172

**Published:** 2025-01-07

**Authors:** Yasunobu Yamashita, Hirofumi Yamazaki, Akiya Nakahata, Tomoya Emori, Yuki Kawaji, Takashi Tamura, Masahiro Itonaga, Reiko Ashida, Masayuki Kitano

**Affiliations:** Second Department of Internal Medicine, Wakayama Medical University, 811-1, Kimiidera, Wakayama 641-0012, Japan; hirofumi.y.nagoya@gmail.com (H.Y.); akiya1108nakahata@gmail.com (A.N.); t-emori@wakayama-med.ac.jp (T.E.); y.kawaji1985@gmail.com (Y.K.); ttakashi@wakayama-med.ac.jp (T.T.); masaitonaga0907@gmail.com (M.I.); rashida@wakayama-med.ac.jp (R.A.); kitano@wakayama-med.ac.jp (M.K.)

**Keywords:** pancreatic lesions, contrast-enhanced EUS, EUS elastography, EUS microvascular imaging, EUS-guided tissue acquisition

## Abstract

Endoscopic ultrasonography (EUS) is one of the most reliable and efficient diagnostic modalities for detecting pancreatic lesions because its spatial resolution is superior to that of other imaging modalities. EUS is now used widely in clinical practice. Moreover, the modality has been developed such that we now have contrast-enhanced EUS, EUS elastography (including strain elastography and shear wave elastography), and EUS microvascular imaging (detective flow imaging), all of which have improved diagnosis of pancreatic lesions. Pathological diagnosis coupled with EUS-guided tissue acquisition (EUS-TA) is superior to other imaging modalities. Indeed, EUS-TA is used not only for diagnosis but also to collect tissue samples for cancer gene panel testing, allowing for a more personalized approach to treatment. Therefore, EUS is an indispensable and important modality in the diagnosis of pancreatic lesions.

## 1. Introduction

Pancreatic cancer is the fourth deadliest cancer in the U.S. and Japan and is increasing [1,2]. According to the American Cancer Society, 50,550 people in the USA died of PC in 2023 [1], and the cancer statistics in Japan showed 39,468 deaths in 2022 [2]. PC has a very poor 5-year survival rate (12% in 2023) [1]; therefore, early detection and differential diagnosis of pancreatic neoplasms [3] are important to improve the prognosis. Moreover, differential diagnosis is important because about 60% of pancreatic tumors smaller than 15 mm are not PC [4]; therefore, appropriate diagnosis can avoid unnecessary radical surgery.

Endoscopic ultrasonography (EUS) is the best diagnostic tool for small pancreatic neoplasms due to high spatial resolution; however, differential diagnosis by EUS alone has limitations because many different pancreatic neoplasms show up as a hypoechoic mass. In this regard, evaluation of the vascularity and tissue elasticity of pancreatic neoplasms are additional methods that improve the ability to characterize pancreatic neoplasms.

Doppler imaging modalities such as color-Doppler EUS, power-Doppler EUS, e-FLOW EUS, and detective flow imaging EUS (DFI-EUS), as well as contrast-enhanced EUS, are used currently for real-time examination of vascularity. These modalities have proven very useful in the diagnostic field. Moreover, they are superior to contrast-enhanced computed tomography (CT) and magnetic resonance imaging (MRI) because they can be used for patients in whom the use of contrast dyes is contraindicated (e.g., patients with renal failure or allergy to the contrast agents). They allow for dynamic and repeat evaluation.

Elasticity is assessed using EUS elastography techniques such as strain elastography and shear wave elastography. Again, these methods are very useful for the diagnosis of PC. Another feature of EUS is the ability to obtain tissue samples; for example, when a pancreatic neoplasm is depicted, a sample can be taken using EUS-guided tissue acquisition (EUS-TA). EUS-TA has made it possible to collect a large number of tissue samples, which is helpful not only for diagnosing cancer but also for determining the course of treatment through genetic testing. This review focuses on advances in EUS-based diagnosis, particularly its utility for the diagnosis of pancreatic lesions.

## 2. Method

### Literature Search and Review

We summarized the results of EUS based diagnosis of pancreatic neoplasms. The PubMed/Medline database was searched before May 2024 using the search. “EUS” or “endoscopic ultrasound” and “pancreatic cancer” or “pancreatic neoplasm”. After a literature search, we conducted a narrative review outlining diagnostic techniques using EUS for pancreatic neoplasms, including the history of EUS diagnosis, as well as the latest diagnostic methods to date.

**I.** 
**Advances in EUS for diagnosis of pancreatic diseases**



**Advances in EUS (**
Figure 1
**)**


**Figure 1 cancers-17-00172-f001:**
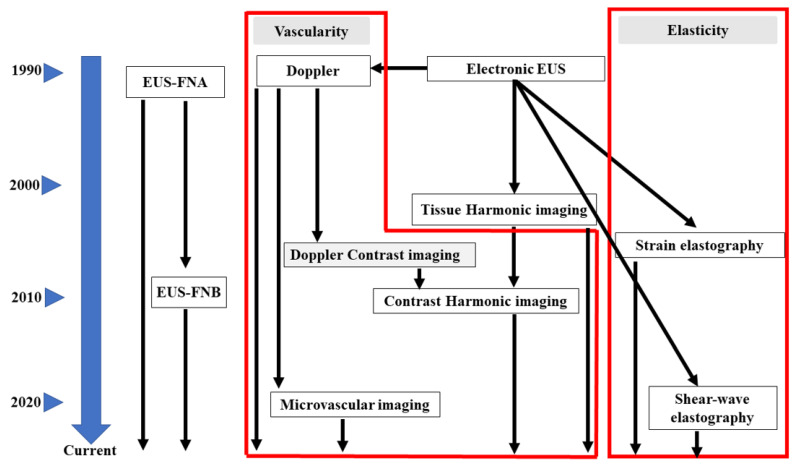
**The time sequence diagram for the development and revolution of EUS equipment.** EUS-FNA, EUS-guided fine needle aspiration; EUS-FNB, EUS-guided fine needle biopsy.

***(i)*** 
**
*B mode imaging*
**


EUS systems fitted with a mechanical radial scanning transducer have been used widely since the 1980s, and they have become useful for the diagnosis of pancreatic neoplasms; however, mechanical radial scanning cannot detect Doppler signals. Electronic scanning technology allows the detection of harmonic components within insonated tissue; these components are the result of nonlinear sound propagation. The development of tissue harmonic EUS during the 2000s markedly improved the resolution of EUS images.

***(ii)*** 
**
*Assessment of vascularity*
**


In 1988, the field of EUS introduced electronic scanning, which allows color or power-Doppler imaging; however, although electronic techniques can detect large vessels, they cannot depict slow flow in fine vessels or perform imaging of parenchymal perfusion. In this regard, Kato et al. performed the first contrast-enhanced EUS (CE-EUS) of a pancreatic lesion with arterial administration of CO_2_ for the celiac or superior mesenteric artery [5]; however, this method is more invasive than conventional EUS due to the need for angiography prior to image capture. In 1999, first-generation contrast agents that can be administered through a peripheral vein prior to transabdominal ultrasonography were introduced, although contrast-enhanced EUS proved difficult because the image is obtained by oscillating and breaking the bubble with high acoustic power; such high power was difficult to achieve using small EUS transducers.

Despite these difficulties, contrast-enhanced Doppler EUS with first-generation contrast agents administered via a peripheral vein increased the sensitivity of signals emitted from vessels; this is because pseudo-Doppler signals are generated from the microbubbles. However, contrast-enhanced Doppler EUS cannot generate an image of parenchymal perfusion and suffers from artifacts such as blooming and overpainting, in which vessels appear larger than they really are (Figure 2). The development of second-generation contrast agents for ultrasonography meant that microbubbles could be oscillated or broken by an EUS transducer of lower acoustic power. In 2008, contrast-enhanced harmonic EUS (CH-EUS) finally succeeded in visualizing perfusion of the microvasculature within the pancreatic parenchyma.

Color-Doppler EUS or power-Doppler EUS are effective for real-time monitoring of vascularity without the need for contrast agents; however, these EUS modes have poor spatial resolution. In 2007, the e-FLOW technique was introduced to improve resolution color or power-Doppler imaging. Nevertheless, when motion artifacts are removed using conventional Doppler techniques, the flow in smaller vessels can no longer be visualized due to the loss of slow flow data. In 2018, microvascular imaging, a recent development in ultrasonography, improved upon conventional Doppler imaging, particularly with respect to poor spatial resolution and low sensitivity for slow flow. Microvascular imaging uses a unique algorithm to remove motion artifacts, which then permits the visualization of fine vessels with slow flow velocities without the need for contrast agents.

***(iii)*** 
**
*Assessment of tissue elasticity*
**


In 2005, a study reported EUS elastography of the pancreas using a strain method, which provides information about the strain made by compression of the target tissue by the EUS probe, or cardiovascular pulsation through the aorta [6]. Although the original technique provided qualitative evaluation of tissue elasticity, there were two semiquantitative methods for the analysis of tissue stiffness in second-generation strain EUS.

The first is a quantitative evaluation method using the strain ratio of reference tissue area to the region of interest.

The second is to create a histogram using the average of the number of strains, with numbers from 0 to 255, and evaluate them quantitatively.

Nevertheless, it is difficult to evaluate strain EUS elastography measurements across patient groups because the technique does not measure absolute values of elasticity. To resolve this, shear wave elastography, which provides absolute values for pancreatic hardness based on measurement of shear wave velocity (Vs), was introduced in 2017.

***(iv)*** 
**
*EUS-guided tissue acquisition (EUS-TA)*
**


EUS using a curved array was developed in 1989; however, in 1992, Vilmann et al. first reported the concept of EUS-guided fine-needle aspiration (EUS-FNA), a method based on electronic scanning conducted using a linear array transducer [7]. Initially, EUS-FNA was used mainly for cytology. In 2002, the Trucut needle was developed to allow more tissue samples to be collected [8]. To improve puncture performance, novel needles (such as reverse bevel needles) were then developed to enable the acquisition of tissue cores using a technique called EUS fine-needle biopsy (FNB). Subsequently, next-generation needles such as Fork-tip and Franseen needles appeared. To date, three different needle sizes are available commercially: 19-, 22-, and 25-gauge.

**II.** 
**Utility of EUS for diagnosis of Pancreatic solid lesion (Table 1)**


**(A)** 
**Conventional EUS**


Table 1 lists the typical findings/features of solid pancreatic lesions. In 1984, Yasuda et al. reported that EUS was significantly more accurate (91%) than CT (66%) and abdominal ultrasonography (64%) for the diagnosis of PC [9]. A meta-analysis of the diagnostic utility of EUS for detecting PC (10 studies, 669 patients in total) was published in 2014; it found that the pooled sensitivities and specificities were 76.7% and 91.7% in EUS, 57.9% and 90.6% in retrograde cholangiopancreatography (ERCP), and 79.9% and 94.2% in EUS plus ERCP, respectively. Thus, rates of PC diagnosis by EUS and EUS plus ERCP are significantly higher than those by ERCP [10]. However, in terms of ERCP, post-ERCP pancreatitis should be particularly noteworthy because it occurs in medical practice and can lead to major complications [11].

Moreover, the pooled sensitivity, pooled specificity, and area under the curve (AUC) of detection of PC with EUS in another metanalysis were 85%, 58%, and 0.8, respectively, when CT was missed [12]. Thus, EUS is a necessary tool for the diagnosis of PC that is not detectable by other modalities, particularly when PC is suspected due to indirect findings, such as a dilation of the main pancreatic duct (Figure 3).

Meta-analysis of 52 studies including 3567 patients showed that diagnosis of PC with EUS, transabdominal US, CT, MRI, and PET was 91%, 88%, 90%, 93%, and 89% in sensitivity, 86%, 94%, 87%, 89%, and 70% in specificity, and 89%, 91%, 89%, 90%, and 84% in accuracy, respectively. There were no significant differences among each modality for the diagnostic performance of PC [13]. However, EUS is a valuable diagnostic tool for early-stage PCs. Kanno et al. reported that rates of detection of pancreatic tumors by abdominal ultrasonography, CT, MRI, and EUS were 8.8%, 10%, 10.9%, and 24.4% for stage 0 tumors, respectively, and 67.3%, 65.8%, 57.5%, and 92.4% for stage I tumors, respectively [14]; therefore, EUS is the best diagnostic tool for early-stage PC. Therefore, EUS is recommended as a diagnostic tool in patients with suspected PC, especially as it is more sensitive than other imaging modalities for the diagnosis of PC in clinical practice guidelines for PC [15]. On the other hand, in clinical practice, patients were divided into three groups of resectable, borderline resectable, and unresectable pancreatic cancer according to contact or no contact with the portal vein (PV)/superior mesenteric vein (SMV), celiac axis, hepatic artery and superior mesenteric artery and contact of ≤180° or contact of >180° with the PV/SMV and contact of ≤180° or contact of >180° with the superior mesenteric artery. Therefore, the diagnosis of vascular invasion is important in determining resectability [16]. Contrast-enhanced CT is the gold standard for determining treatment options, including indication for surgery, determination of the need for neoadjuvant chemotherapy (NAC), and assessment after NAC [17]. Therefore, CT should be used in combination with EUS to determine these.

With respect to neuroendocrine neoplasms (NENs), detection rates by EUS (94.5%) are significantly superior to those by CT (86.3%). Moreover, EUS is significantly superior to CT with respect to the detection of PNENs ≤ 5 mm (58.3% vs. 16.7%, respectively) and 5–10 mm (97.7% vs. 79.5%, respectively). By contrast, there is no significant difference in detection rates between EUS and CT with respect to NENs > 10 mm (98.4% vs. 96.4%, respectively) [18]. A meta-analysis of preoperative detection of NENs (17 studies, 612 patients) revealed that EUS identified NENs in 97% of cases, and all studies reported improved detection of NENs when using EUS [19]; therefore, EUS is particularly important for the diagnosis of small NENs.

**(B)** 
**Doppler EUS**


The sensitivity, specificity, and accuracy of PC detection by EUS is 93%, 77%, and 88%, respectively, when the absence of power Doppler signals is defined as PC [20].

The sensitivity CT, power-Doppler EUS, and Doppler contrast EUS using a first-generation contrast agent for diagnosis of PCs of ≤2 cm is 50.0%, 11.0% and 83.3%, respectively; thus, Doppler contrast EUS is significantly more sensitive than power-Doppler EUS and CT [21]. However, problems such as no parenchymal perfusion image, blooming, overpainting, poor spatial resolution, and low sensitivity for slow flow persist.

**(C)** 
**CH-EUS**


In CH-EUS, pancreatic cancers are typically depicted as hypovascular, inflammatory masses as isovascular, and NETs as hypervascular compared with the surrounding pancreatic tissues. In the previous report, 104 of 109 (94%) PCs were hypovascular, eight of 11 (72%) were inflammatory masses, eight of nine (89%) autoimmune pancreatitis were isovascular, and five of eight (63%) NETs were hypervascular (Figure 4) [22]. They compared the pathology image to the early phase contrast pattern and showed that the pathology in the hypovascular pattern showed heterogeneous tumor cells, necrotic tissue, fibrous tissue, and few vessels, and in the isovascular pattern, homogeneous tumor cells, abundant vessels, and no necrotic or fibrous tissue were found.

In a meta-analysis comparing CH-EUS in 719 patients and conventional EUS in 723 patients, the sensitivity, specificity, diagnostic odds ratio, and AUC were 93%, 80%, 57.9, and 0.96 for CH-EUS and 86%, 59%, 8.3, and 0.80 for conventional EUS. For the diagnosis of PC, CH-EUS was 2.98 times more accurate than conventional EUS [23].

For small PCs (11–20 mm), the sensitivity, specificity, and accuracy of CH-EUS, CT, and MRI are 95%, 83%, and 94%; 78%, 83%, and 79%; and 73%, 33%, and 68%, respectively; therefore, the diagnostic ability of CH-EUS is significantly superior to that of CT and MRI. Moreover, the sensitivity, specificity, and accuracy of CH-EUS, MDCT, and MRI for the detection of PC ≤ 10 mm is 70%, 100%, and 77%; 20%, 100%, and 38%; and 50%, 100%, and 62%, respectively [24]. Therefore, CH-EUS is indispensable for the diagnosis of PCs.

In terms of malignant potential, NENs are considered to be aggressive if morphologic/histologic findings are consistent with metastatic disease (i.e., adjacent organ involvement, lymph node involvement, distant organ metastasis), and/or histologic findings suggest a G3 tumor (Ki67 > 20%). The diagnostic ability for tumor aggressiveness with CH-EUS was 96% in sensitivity, 82% in specificity, and 86% in accuracy, respectively [25]. Therefore, CH-EUS contributes not only to aid diagnosis but also enables the estimation of the malignant potential of NENs.

**(D)** 
**Strain elastography**


Seventy-seven PCs, 42 inflammatory masses, and 10 NENs were classified into categories using the homogeneity of the color map (homogeneous or heterogeneous) and the combination of the dominant color within a color map (blue or green). They defined the combination of blue and heterogeneous as PC (Figure 5), green and heterogeneous as a benign inflammatory mass, and blue and homogeneous as a NEN. The diagnostic ability for malignant pancreatic solid lesions was 100% in sensitivity and 85.5% in specificity, respectively [26]. In a meta-analysis of 1687 patients, the diagnostic capabilities of qualitative EUS and quantitative EUS elastography for malignant pancreatic lesions were 0.98 and 0.95 in sensitivity and 0.63 and 0.61 in specificity, respectively [27]. Therefore, EUS strain elastography is an important tool for PC.

**(E)** 
**Shear wave elastography**


The shear wave velocity (Vs, m/s) is calculated between two search points with a trackable pulse. If the tissue is hard, a faster shear wave propagates. Because the technique is so new, there is only one report reporting the detection of a pancreatic lesion using shear wave elastography. The median Vs (m/s) values for solid pancreatic lesions were as follows: 2.19 for PC, 1.31 for NENs, 2.56 for inflammatory masses, and 1.58 for metastatic tumors. There was no significant difference between the Vs for these diseases [28]. The authors concluded that EUS shear wave tended to be unstable when used to measure the elasticity of solid pancreatic lesions; therefore, additional studies should investigate the utility and indications for EUS shear wave as a method of detecting solid pancreatic neoplasms.

**(F)** 
**EUS-TA**


A meta-analysis of EUS-FNA and EUS-guided fine needle biopsy (EUS-FNB) for the diagnosis of PC (18 articles; 2695 patients) revealed that the pooled diagnostic accuracy and tissue core sampling rate for EUS-FNB (87% and 80%, respectively) were significantly higher than those for EUS-FNA (80% and 62%, respectively); in addition, EUS-FNB required significantly fewer needle passes for diagnosis [29].

A meta-analysis of the best needle size for diagnosis (eight articles; 1292 patients) revealed that the pooled sensitivity and specificity of a 22G needle were 0.85 and 1, respectively, whereas those for a 25G needle were 0.93 and 0.97, respectively; therefore, the 25G needle results in significantly higher sensitivity [30].

A multicenter study of 13,566 EUS-FNA cases reported adverse events in 234 patients (1.7%). Pancreatitis and bleeding accounted for 26.5% and 49.1% of all adverse events, respectively [31].

In a network meta-analysis including 756 patients for the EUS-TA sampling technique, sample adequacy with the no-suction technique was significantly inferior to the other techniques (RR, 0.85 [95% CI, 0.78–0.92] vs. slow pull; RR, 0.85 [95% CI, 0.78–0.92] vs. dry suction; RR, 0.83 [95% CI, 0.76–0.90] vs. wet suction). Tissue integrity with the wet-suction technique was significantly superior to dry suction significantly in terms of tissue integrity of the sample (RR, 1.36; 95% CI, 1.06–1.75). Therefore, wet suction is the best technique for EUS-TA [32]. In terms of sample adequacy for genomic profiling, EUS-FNB was significantly superior to EUS-FNA for mean DNA concentrations (5.930 μg/mL vs. 3.365 μg/mL, respectively) [33]. In over 95% of PC patients, KRAS, TP53, CDKN2A, and SMAD4 were identified as major driver genes for PC mutational activation of the KRAS oncogene [34]. Therefore, next-generation sequencing (NGS) using the KRAS mutation as a reference has been validated. KRAS mutations were found in 96% (26/27) of PC samples and 0% (0/11) of non-PC samples by EUS-TA [35]. Regarding the adequacy of NGS for PC with EUS-TA samples, 70.1% of adequate samples were obtained with 167 samples, including 145 EUS-FNA and 22 EUS-FNB. EUS-FNB (90.9%) was superior to EUS-FNA (66.9%) in terms of adequate samples for NGS [36].

**(G)** 
**EUS-FNA combined with CH-EUS or EUS elastography**


A meta-analysis of six articles, including 701 patients, compared the diagnostic utility of CH-EUS-FNA and EUS-FNA for PC. The pooled diagnostic sensitivity of CH-EUS-FNA was 84.6%, and that of EUS-FNA was 75.3%; therefore, CH-EUS-FNA was significantly superior (odds ratio (OR), 1.74) for PC diagnosis. Moreover, pooled sample adequacy was 95.1% (91.1–99.1%) for CH-EUS-FNA (95.1%), significantly superior to that of EUS-FNA (89.4%), with an OR of 2.40 [37].

Kongkam et al. reported the sensitivity, specificity, positive predictive value, negative predictive value, and accuracy of EUS-FNA alone versus EUS-FNA combination with EUS elastography by strain ratio and found that they were 90%, 100%, 100%, 80%, and 92.9%, respectively, versus 95.2%, 71.4%, 90.9%, 83.3%, and 89.3%, respectively [38]. Thus, the combination of EUS-FNA and EUS elastography by strain ratio was not superior to EUS-FNA alone.

**(H)** 
**CH-EUS for the treatment efficacy of endoscopic radiofrequency ablation (RFA)**


After the RFA session, treatment efficacy was confirmed by the disappearance of intratumoral enhancement on CH-EUS. Therefore, CH-EUS may be useful in assessing treatment response after RFA and residual viable tumor during additional RFA [39].

**III.** 
**Utility of EUS for diagnosis of pancreatic cysts**


**(A)** 
**Conventional EUS**


Typical pancreatic cyst findings with respect to the appearance, internal structure, main pancreatic duct (MPD), and MPD communication are Grape-like/cyst by cyst/dilation/communication in IPMN), Orange-like cyst in cyst/normal/infrequent/non-communication in mucinous cyst neoplasms, and spongy or honeycomb-like/micro and (or) macro-cystic/normal/non-communication in serous cyst neoplasm, respectively.

One study reported that in 102 BD-IPMN follow-up patients, the sensitivity/specificity of EUS, abdominal ultrasonography, CT, and MRI for detection of PC in IPMN patients were 100%/100%, 39%/99%, 56%/100%, and 50%/100%, respectively; therefore, EUS is significantly superior to the other three imaging modalities for diagnosis of IPMN-related PC [40].

**(B)** 
**CH-EUS**


According to guidelines for IPMN [41], the presence or absence of mural nodules is a key factor in the decision to perform surgery; accurate evaluation can be problematic. Sometimes, they struggle to differentiate mural nodules from mucous clots. CH-EUS can be useful for assessing vascularity in mural lesions, which is helpful for differential diagnosis of mural lesions (Figure 6). The sensitivity, specificity, positive predictive value, negative predictive value, and accuracy of CH-EUS for detecting mural nodules is 100%, 80%, 92%, 100%, and 94%, respectively, whereas those for CT are 58%, 100%, 100%, 50%, and 71%, respectively [42]. With respect to the differential diagnosis of benign and malignant IPMNs, a meta-analysis of eight studies involving 320 patients with CH-EUS revealed that the pooled sensitivity, specificity, and diagnostic accuracy for the diagnosis of high-grade dysplasia (HGD) or invasive carcinoma (IC) within mural nodules is 97.0%, 90.4%, and 95.6%, respectively, when mural lesion with vascularity was defined as the malignancy [43]. Therefore, guidelines recommend CH-EUS in cases of IPMN with suspicion of HGD/IC if the clinical setting is available for the assessment of high-risk stigmata (HRS) and worrisome features (WFs) [41].

**(C)** 
**DFI-EUS**


DFI-EUS (32/33 cases, 97%) is significantly superior to e-FLOW EUS (24/33 cases, 73%) for differential diagnosis between mural nodules and mucous clots in IPMN and between solid gallbladder lesions and gallbladder sludge (Figure 7) [44]. Although no clear conclusion can be drawn due to including gallbladder disease in this study, DFI-EUS is expected to be a useful tool for differential diagnosis between mural nodules and mucous clots in IPMN and will likely replace CH-EUS in the future.

**(D)** 
**EUS-TA**


With respect to the use of tumor markers in pancreatic fluid for the diagnosis of IPMN by EUS-FNA, the sensitivity of the modality for a CEA level > 200 ng/mL and a CA 72-4 level > 40 U/mL was 44% and 39%, respectively. Moreover, the diagnostic performance for malignant IPMNs with a CEA level > 200 ng/mL was 90% for sensitivity, 71% for specificity, 50% for positive predictive value, and 96% for negative predictive value [45]. However, in a meta-analysis of 609 lesions for differential diagnosis between mucinous and non-mucinous cystic neoplasms, the measurement of pancreatic cyst fluid glucose levels (91%) was significantly superior to CEA measurements (56%) in pooled sensitivity [46]. In a meta-analysis of the differential diagnosis between benign and malignant IPMN, EUS-FNA-based cytology had a pooled sensitivity of 64.8% and a pooled specificity of 90.6% [47]. In a meta-analysis including 575 patients for differential diagnosis between neoplastic and non-neoplastic pancreatic cystic lesions, EUS-guided through-the-needle biopsy (EUS-TTNB) was 76.60% in pooled sensitivity, 98.90% in pooled specificity (95% CI = 93.80–100.00), and 41.34 in pooled diagnostic odds ratio [48]. Therefore, EUS-TTNB has good sensitivity and excellent specificity for diagnosis. In terms of adverse events, in a multicentre retrospective analysis, there were adverse events in 58 patients (11.5%). Multivariate analysis showed that age, number of TTNB procedures, complete aspiration of the cyst, and diagnosis of IPMN were independent predictors [49]. EUS-guided needle-based confocal laser endomicroscopy (nCLE) is a new imaging technique. After passing a thin mini-probe through a 19G needle, in vivo images of the pancreas can be obtained at the cellular level. If a villous structure (finger-like projections) is seen in a pancreatic cyst, the cyst is diagnosed as an IPMN. If a superficial vascular network is seen, the cyst is diagnosed as a serous cystadenoma (SCA). The specificity for IPMN and SCA on nCLE images was 100% in articles [50,51,52]. The sensitivity and specificity of EUS-TA for diagnosis of mucinous cyst or IPMN with a GNAS and/or KRAS mutation are 83.7% and 81.8%, respectively, and 87.2% and 84.6%, respectively [53].

**IV.** 
**Future of echoendoscopy**


Although EUS allows detailed observations, the utility of diagnosis is endosonographer-dependent. Under such circumstances, artificial intelligence (AI) is being assessed as a better, more reliable method of interpretation. A total of 1497 EUS images, including malignancies, neuroendocrine tumors, benign cysts, chronic and acute pancreatitis, normal fatty pancreas, and benign lesions, were analyzed in 165 patients. The best AI model demonstrated detection and segmentation of the test set, with a mean intersection over the union of 0.73 and a PPV, NPV, total accuracy, and ROC of 0.82, 0.96, 0.95, and 0.95, respectively [54]. This provides the basis for a computer-aided detection system for EUS, which would be valuable for future detection and evaluation of pancreatic lesions.

This study has some limitations. This was a narrative review, not a systematic review, and there was a selection bias in the literature.

## 3. Conclusions

Conventional EUS is a key method used to identify pancreatic neoplasms; however, differential diagnosis of pancreatic lesions remains challenging. This situation has been improved markedly by development of EUS-TA, Doppler EUS, CH-EUS, and EUS elastography. Moreover, EUS-TA provides the benefit of precision medicine-based approach. Advances in EUS mean that the modality has become indispensable for diagnosis of pancreatic neoplasms.

## Figures and Tables

**Figure 2 cancers-17-00172-f002:**
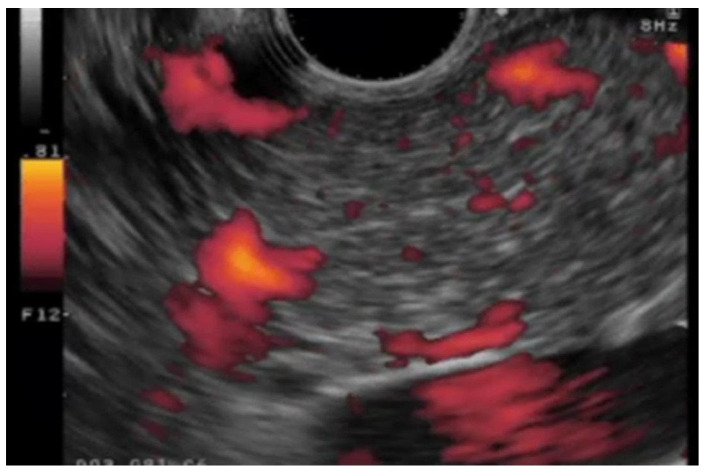
**Doppler EUS.** Doppler EUS image showing artifacts such as blooming and overpainting, in which vessels appear larger than they really are.

**Figure 3 cancers-17-00172-f003:**
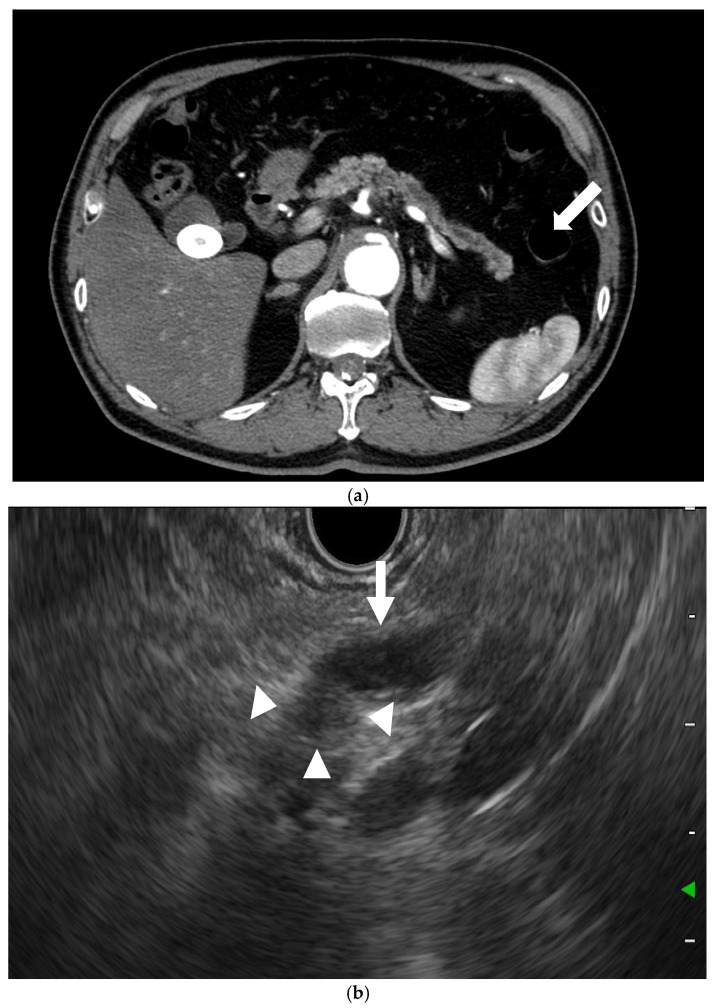
**EUS for detection of pancreatic cancers missed by CT.** (**a**) CT: The main pancreatic duct dilation (arrow) was dilated, but no pancreatic lesion was detected. (**b**) EUS: A small hypoechoic lesion (9 mm; arrowhead), with dilation of the main pancreatic duct (arrow), was detected.

**Figure 4 cancers-17-00172-f004:**
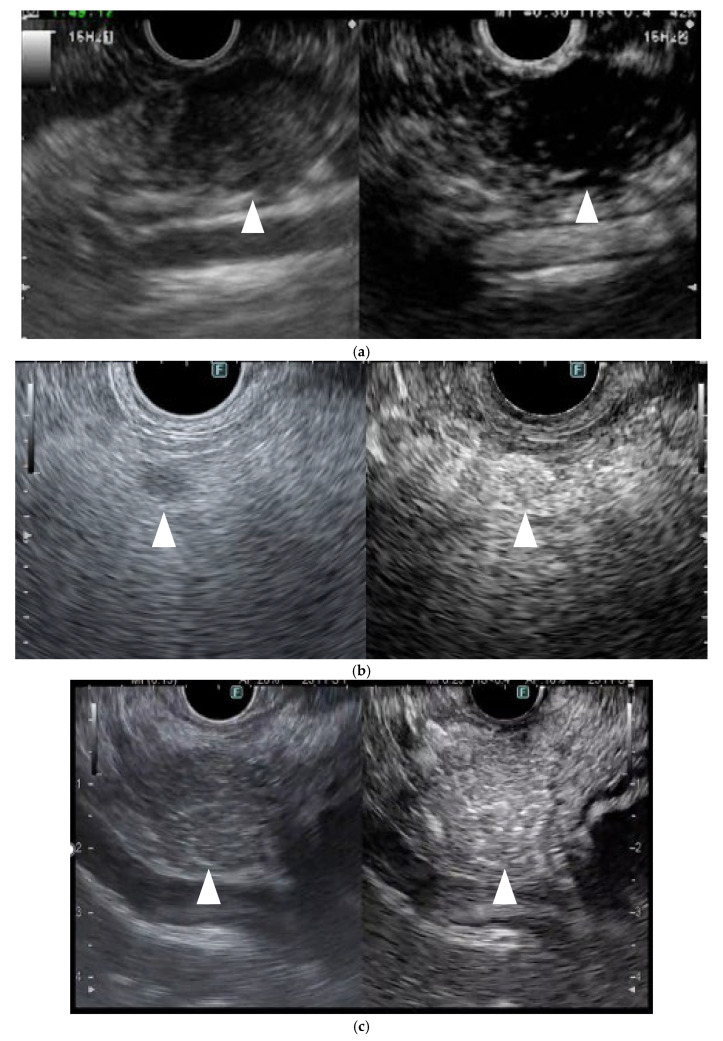
**Typical contrast-enhanced harmonic EUS (CH-EUS) images of a pancreatic lesion.** (**a**) Representative example of a pancreatic cancer. (Left) B mode EUS detected a pancreatic lesion as an irregular, heterogeneous, and hypoechoic lesion (arrowhead). (Right) CH-EUS shows the pancreatic lesion (arrowhead) as hypovascular compared with the surrounding pancreatic tissue. (**b**) Representative example of a neuroendocrine neoplasm. (Left) B mode EUS detected a round hypoechoic lesion (arrowhead) pancreatic lesion. (Right) CH-EUS showed the pancreatic lesion to be hypervascular (arrowhead) compared with the surrounding pancreatic tissue. (**c**) Representative example of an inflammatory mass. (Left) B mode EUS detected a pancreatic lesion as a low echoic mass (arrowhead). (Right) CH-EUS showed that the pancreatic lesion was isovascular (arrowhead) compared with the surrounding pancreatic tissue.

**Figure 5 cancers-17-00172-f005:**
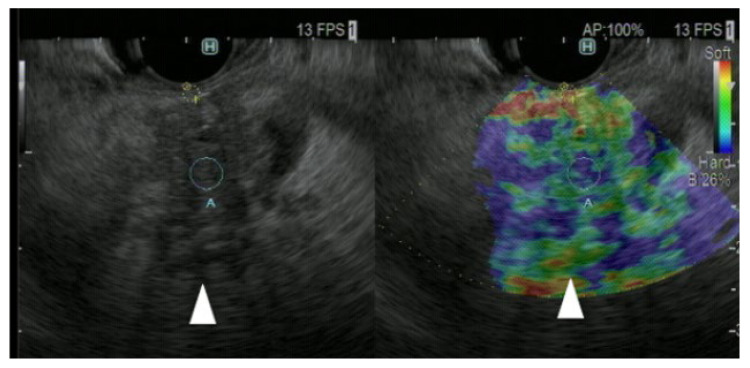
**EUS strain elastography images of pancreatic cancer.** Pancreatic lesion was detected as a low echoic lesion (arrowhead) on conventional EUS (**left**). EUS strain elastography detected a pancreatic lesion with a heterogeneous blue pattern (arrowhead) compared with that of surrounding pancreatic tissue (**right**).

**Figure 6 cancers-17-00172-f006:**
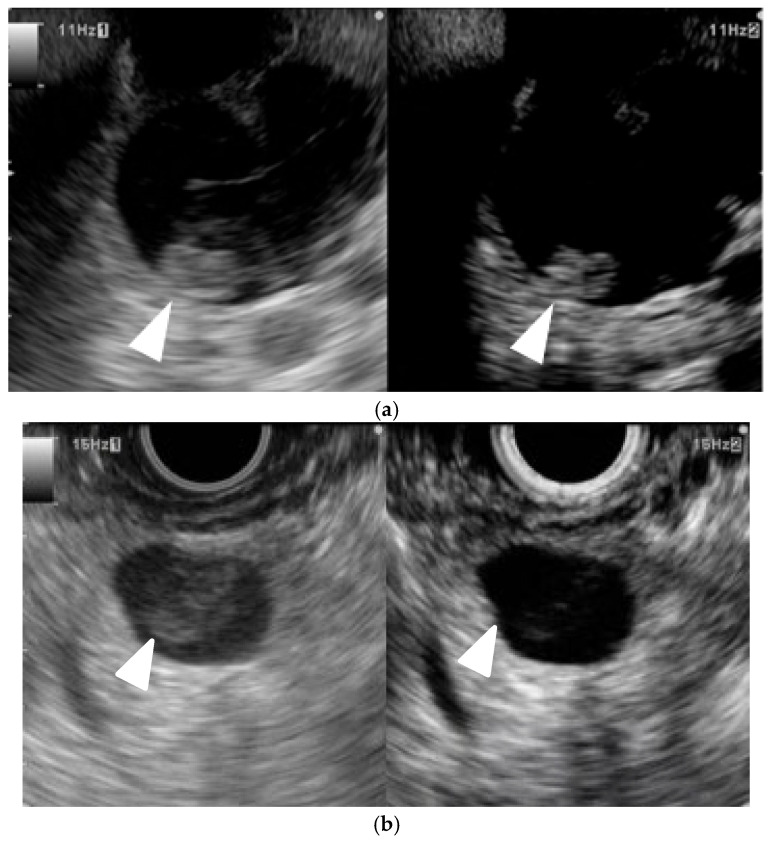
**Typical contrast-enhanced harmonic EUS (CH-EUS) images of a mural lesion in IPMN.** (**a**) Mural nodule. B mode EUS (left) shows an isoechoic mural lesion (arrowhead) in a cyst. CH-EUS (right) shows a mural lesion with vascularity (arrowhead). (**b**) Mucous clot. B mode EUS (left) shows an isoechoic mural lesion (arrowhead) in a cyst. CH-EUS (right) shows a mural lesion without vascularity (arrowhead).

**Figure 7 cancers-17-00172-f007:**
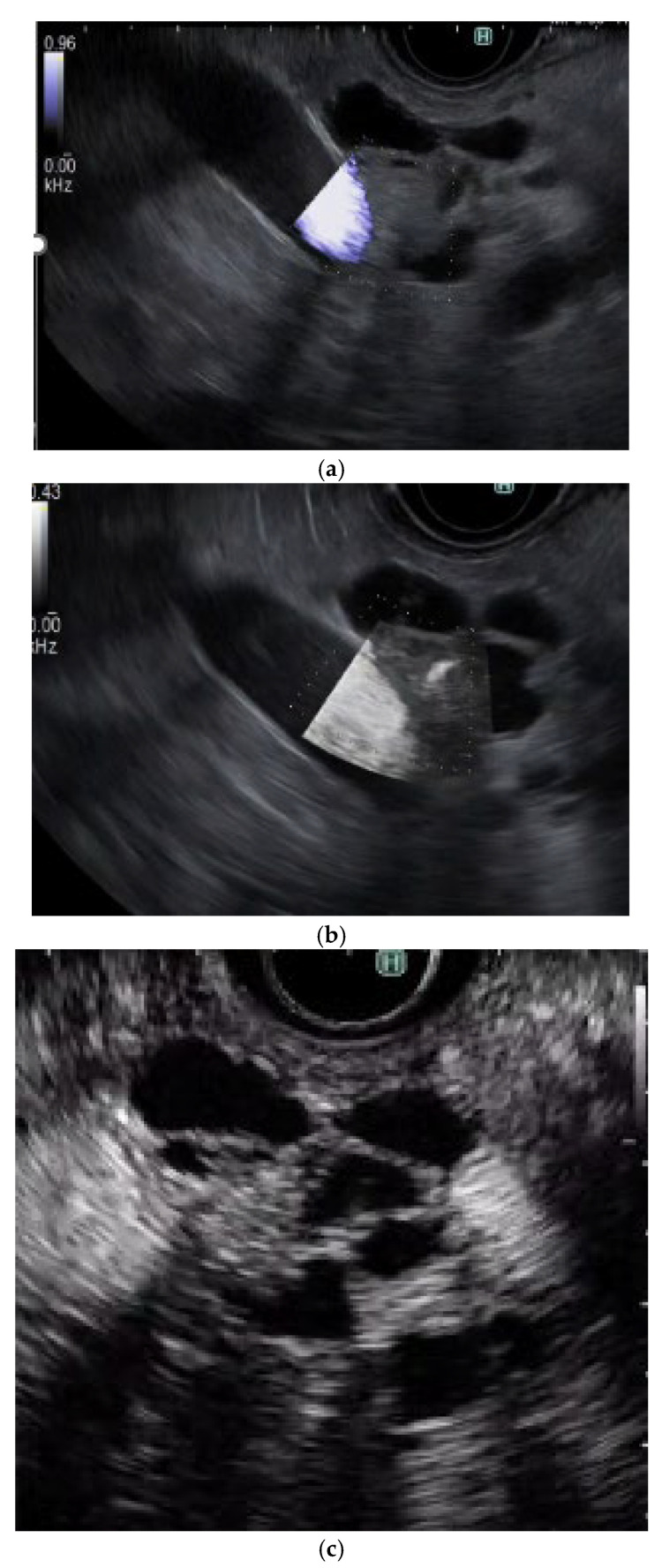
**The sensitivity of vascularity for detecting a mural nodule in IPMN using each modality.** (**a**) e-Flow EUS: No vessels are detected in the mural nodule. (**b**) DFI-EUS: Detected a vessel in the mural nodule. (**c**) CH-EUS: Detected a mural nodule with vascularity.

**Table 1 cancers-17-00172-t001:** Typical conventional EUS findings with respect to solid pancreatic neoplasms.

	Pancreatic Cancer	Inflammatory Mass	Neuroendocrine Neoplasm
B mode	Heterogeneous	Heterogeneous	Homogeneous, but heterogeneous (malignant)
	Hypoechoic	Calcification	Round
	Irregular margin	Cysts	Hypoechoic
		Peripancreatic echo-rich stranding	Clearly demarcated
Contrast-enhanced	Hypovascular	Isovascular	Hypervascular
Elastography (strain)	Heterogeneous blue pattern	Green to blue pattern according to the degree of fibrosis	Homogeneous blue pattern

## Data Availability

Not applicable.

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
