# Peer review of "Advances in Endoscopic Ultrasonography-Based Diagnosis of Pancreatic Lesions: Narrative Review"

_cancers, 2025, doi:10.3390/cancers17020172_

Round 1
Reviewer 1 Report
Comments and Suggestions for Authors
I have read with interest the detailed review on the advances of EUS in pancreatic cancer.
The review is well written and very interesting. However, some more useful data is missing. In the section on pancreatic cysts there is no mention of the role of glucose determination in distinguishing between mucinous and non-mucinous cysts. There is scientific evidence that glucose determination is superior to CEA in this regard.
On the other hand, there is also no mention of the role of intracystic biopsy in obtaining histology, especially of mural nodules.
Finally, a small section on the future of echoendoscopy and the role of artificial intelligence could be useful.
Author Response
We thank the Reviewer for the helpful and constructive comments, which have helped to improve the manuscript.
Major
1." In the section on pancreatic cysts there is no mention of the role of glucose determination in distinguishing between mucinous and non-mucinous cysts. There is scientific evidence that glucose determination is superior to CEA in this regard."
Authors' reply: Thank you for your informative comment. As requested, we have now mentioned the role of glucose determination in distinguishing mucinous from non-mucinous cysts:
“a meta-analysis conducted to differentiate mucinous from non-mucinous cystic neoplasms (eight studies, including 609 lesions) revealed that the pooled sensitivity of EUS-guided pancreatic sampling to measure pancreatic cyst fluid glucose levels in IPMN was significantly higher than for CEA measurements (91% [95% confidence interval {CI}, 88–94; I2 = .00] vs. 56% [95% CI, 46–66; I2 = 537.14]; P < .001)” (page 15, lines 386-391).
2." there is also no mention of the role of intracystic biopsy in obtaining histology, especially of mural nodules."
Authors' reply: Thank you for pointing this out. We have now mentioned the role of intracystic biopsy in obtaining samples for histology:
“With respect to biopsy methods based on EUS-FNA, a meta-analysis of 11 studies (including 575 patients) revealed that the pooled sensitivity and specificity of EUS-guided through-the-needle biopsy (EUS-TTNB) for distinguishing neoplastic from non-neoplastic pancreatic cystic lesions (PCLs) were 76.60% (95% CI = 72.60–80.30) and 98.90% (95% CI = 93.80–100.00), respectively. The pooled diagnostic odds ratio for EUS-TTNB for diagnosing malignant/premalignant versus non-malignant PCLs was 41.34 (95% CI = 17.42–98.08). EUS-TTNB has good sensitivity, with excellent specificity for accurate classification of PCLs as neoplastic or non-neoplastic. Adding EUS-TTNB to EUS-FNA increases the accuracy of EUS-guided approaches for diagnosing PCLs. A multicenter retrospective analysis of 506 patients with PCLs who underwent TTNB identified adverse events in 58 patients (11.5%). Multivariate analysis identified age (odds ratio [OR] 1.32, 1.09–2.14; p = 0.05), number of TTNB passes (OR increasing from 2.17, 1.32–4.34 to 3.16, 2.03–6.34 as the number of passes increased), complete aspiration of the cyst (OR 0.56, 0.31–0.95; p = 0.02), and diagnosis of IPMN (OR 4.16, 2.27–7.69; p < 0.001) as independent predictors. ” (Page 15, lines 397-411).
3." Finally, a small section on the future of echoendoscopy and the role of artificial intelligence could be useful."
Authors' reply: Thank you for your comment. As requested, we have now mentioned echoendoscopy and the role of artificial intelligence:
“Although EUS allows detailed observations, utility for diagnosis is endosonographer-dependent. Under such circumstances, artificial intelligence (AI) is being assessed as a better, more reliable method of interpretation. A total of 1497 EUS images, including malignancies, neuroendocrine tumors, benign cysts, chronic and acute pancreatitis, normal fatty pancreas, and benign lesions were analyzed in 165 patients. The best AI model demonstrated detection and segmentation of the test set, with a mean intersection over union of 0.73, and an a PPV, NPV, total accuracy, and ROC of 0.82, 0.96, 0.95, and 0.95, respectively. This provides the basis for a computer-aided detection system for EUS, which would be valuable for future detection and evaluation of pancreatic lesions.” (page 15, lines 424-432).
Reviewer 2 Report
Comments and Suggestions for Authors
An excellent well written article outlining the current status of EUS in the diagnosis of pancreatic neoplasms
The only criticism I have is that there is no section which outlines any of its limitations such as the inferiority to other modalities in the assessment of borderline resectable tumours. Addition of a short segment on the limitation of the technique and modality would make the paper more comprehensive in its review
Author Response
We thank the Reviewer for the helpful and constructive comments, which have helped to improve the manuscript.
Major
- " The only criticism I have is that there is no section which outlines any of its limitations such as the inferiority to other modalities in the assessment of borderline resectable tumours. Addition of a short segment on the limitation of the technique and modality would make the paper more comprehensive in its review"
Authors' reply: Thank you for your important comment. We have mentioned some limitations, including inferiority to other modalities with respect to assessment of borderline resectable tumors:
“EUS is less useful for diagnosis of distant metastasis, so CT should be used in combination with EUS to determine the resectability of pancreatic cancer.” (Page 8, lines 191-192).
Reviewer 3 Report
Comments and Suggestions for Authors
the article needs some improvements. THe section about pancreatic cystic lesions is too short and it should be enriched. The authors should comment about the needle confocal laser endomicroscopy and about the through-the-needle biopsy (on this regard, cite the recent series PMID: 35451041)
THe authors should comment on the different sampling techniques for tissue sampling of solid pancreatic masses (see the recent NMA: PMID: 36657607 )
Author Response
We thank the Reviewer for this helpful and constructive comments, which have helped to improve the manuscript.
Major
- " the article needs some improvements. The section about pancreatic cystic lesions is too short and it should be enriched. The authors should comment about the needle confocal laser endomicroscopy and about the through-the-needle biopsy (on this regard, cite the recent series PMID: 35451041)"
Authors' reply: Thank you for your important comments. As requested, we have expanded the text to include mention of needle confocal laser endomicroscopy and through-the-needle biopsy:
“With respect to biopsy methods based on EUS-FNA, a meta-analysis of 11 studies (including 575 patients) revealed that the pooled sensitivity and specificity of EUS-guided through-the-needle biopsy (EUS-TTNB) for distinguishing neoplastic from non-neoplastic pancreatic cystic lesions (PCLs) were 76.60% (95% CI = 72.60–80.30) and 98.90% (95% CI = 93.80–100.00), respectively. The pooled diagnostic odds ratio for EUS-TTNB for diagnosing malignant/premalignant versus non-malignant PCLs was 41.34 (95% CI = 17.42–98.08) [46]. EUS-TTNB has good sensitivity, with excellent specificity for accurate classification of PCLs as neoplastic or non-neoplastic. Adding EUS-TTNB to EUS-FNA increases the accuracy of EUS-guided approaches for diagnosing PCLs. A multicenter retrospective analysis of 506 patients with PCLs who underwent TTNB identified adverse events in 58 patients (11.5%). Multivariate analysis identified age (odds ratio [OR] 1.32, 1.09–2.14; p = 0.05), number of TTNB passes (OR increasing from 2.17, 1.32–4.34 to 3.16, 2.03–6.34 as the number of passes increased), complete aspiration of the cyst (OR 0.56, 0.31–0.95; p = 0.02), and diagnosis of IPMN (OR 4.16, 2.27–7.69; p < 0.001) as independent predictors.
Based on EUS-FNA, EUS-guided needle-based confocal laser endomicroscopy (nCLE) is a new imaging technique that uses a miniprobe thin enough to be passed through a 19G needle. It provides in vivo images of the pancreas at a cellular level, offering the possibility to assess any changes that might have occurred. When a pancreatic cyst has a villous structure (finger-like projections), the lesion is diagnosed as an IPMN. When a superficial vascular network is observed (which is seen only in one type of pancreatic lesion), the lesion is diagnosed as a serous cystadenoma (SCA). Multiple clinical studies have identified pathognomonic signs with a specificity of 100% for IPMN and SCA, which can be seen clearly on nCLE images.” (Page 15, lines 397-420).
- " The authors should comment on the different sampling techniques for tissue sampling of solid pancreatic masses (see the recent NMA: PMID: 36657607 )"
Authors' reply: Thank you. We have now mentioned different techniques for obtaining tissue samples from solid pancreatic masses:
“Different sampling techniques are used for EUS-TA of solid pancreatic lesions. A network meta-analysis of nine randomized controlled trials (including 756 patients) that assessed different techniques for EUS-guided fine-needle biopsy sampling of solid pancreatic masses revealed that the no-suction technique was significantly inferior to the other techniques (RR, 0.85 [95% CI, .78–.92] vs. slow pull; RR, 0.85 [95% CI, .78–.92] vs. dry suction; RR, 0.83 [95% CI, .76–.90] vs. wet suction) in terms of sample adequacy. The wet-suction technique outperformed dry suction significantly in terms of tissue integrity of the sample (RR, 1.36; 95% CI, 1.06–1.75). Therefore, wet suction is the best technique for EUS-TA” (Page 11, lines 300-308).
Round 2
Reviewer 2 Report
Comments and Suggestions for Authors
Although the authors have improved the article, they appear to have misunderstood the thrust of my point. The limitation of EUS in the assessment of borderline resectable tumours (ie those with vascular, particularly arterial, involvement) is not mentioned, nor is there mention of the difficulties in assessment after neoadjuvant therapy. These are only small points but would make the article more complete in its comprehensive assessment of the role of EUS
Author Response
We thank the Reviewer for the helpful and constructive comments, which have helped to improve the manuscript.
" Although the authors have improved the article, they appear to have misunderstood the thrust of my point. The limitation of EUS in the assessment of borderline resectable tumours (ie those with vascular, particularly arterial, involvement) is not mentioned, nor is there mention of the difficulties in assessment after neoadjuvant therapy. These are only small points but would make the article more complete in its comprehensive assessment of the role of EUS"
Authors' reply: We apologize for misunderstanding of your comments. We have mentioned some limitations for assessment of borderline resectable tumors and assessment after neoadjuvant therapy.:
“In clinical practice, patients were divided into three groups of resectable, borderline resectable and unresectable pancreatic cancer according to contact or no contact with the portal vein (PV)/superior mesenteric vein (SMV), celiac axis, hepatic artery and superior mesenteric artery and contact of ≤ 180º or contact of > 180º with the PV/SMV and contact of ≤ 180º or contact of > 180º with the superior mesenteric artery. Therefore, the diagnosis of vascular invasion is important in determining resectability. Contrast-enhanced CT is the gold standard for determining treatment options, including indication for surgery, determination of the need for neoadjuvant chemotherapy (NAC), and assessment after NAC. Therefore, CT should be used in combination with EUS to determine these. ” (Page 8, lines 182-191).
Reviewer 3 Report
Comments and Suggestions for Authors
The manuscript is OK
Author Response
Thank you for your careful peer review.